# SCALABLE ESTIMATION VIA LSH SAMPLERS (LSS)

**Ryan Spring, Anshumali Shrivastava**
Rice University
Houston, TX, USA
{rdspring1, anshumali}@rice.edu

## ABSTRACT

The softmax function has multiple applications in large-scale machine learning. However, calculating the partition function is a major bottleneck for large state spaces. In this paper, we propose a new sampling scheme using locality-sensitive hashing (LSH) and an unbiased estimator that approximates the partition function accurately in sub-linear time. The samples are correlated and unnormalized, but the derived estimator is unbiased. We demonstrate the significant advantages of our proposal by comparing the speed and accuracy of LSH-Based Samplers (LSS) against other state-of-the-art estimation techniques.

## 1 INTRODUCTION

Context-Based Attention is quickly becoming an integral component in deep learning because it allows the model to focus only on specific parts of the data depending on the query. (Vaswani et al., 2017; Olah & Carter, 2016) A classic example of attention is Language Translation with Seq2Seq networks (Sutskever et al., 2014; Bahdanau et al., 2014; Luong et al., 2015). Instead of compressing an entire sequence into a single vector, the encoder generates a series of outputs. The attention mechanism generates a weighted sum of the encoder output sequence for each decoder step using the Softmax function. Unfortunately, the cost associated with content-based attention scales linearly with the length of the sequence, limiting its scalability. In addition, it relies on the less efficient, batch matrix multiplication operation, because the softmax weights change w.r.t the encoding/decoding sequence.

One option is to use approximate nearest-neighbor (ANN) search to efficiently find the top-k elements in the encoding sequence. However, ANN algorithms cannot estimate the softmax weights associated with the top-k elements because of the partition function that normalizes the softmax distribution. In this work, we demonstrate that Locality-Sensitive Hashing (LSH), which is traditionally used for ANN search, is also an efficient, adaptive sampler. We can use the items sampled from the LSH tables to find an accurate, unbiased estimate of the partition function. Therefore, we can use LSH to reduce the cost of softmax attention to sub-linear time.

Our proposed algorithm utilizes LSH hash tables to generate a large set of samples from the proposal distribution $P_{MIPS}(y)$ in near-constant time. This work is an auspicious example of using an algorithmic data structure for efficient and accurate statistical estimation. The obtained samples are correlated, unnormalized, and unlike any known sampling process in the literature. We further show that this unusual property is not a hurdle, and that there exists a simple, unbiased estimator of the partition function using these samples. This proposed algorithm opens a new direction for sampling and unbiased estimation beyond classical IS. We leverage the striking utility of two-decades of LSH/MIPS research for statistical estimation tasks.

## 2 KEY OBSERVATION: LSH IS AN EFFICIENT, INFORMATIVE SAMPLER

The traditional LSH algorithm retrieves a subset of potential candidates for a given query in sub-linear time. For each neighbor in this candidate subset, we compute its actual distance to the query and then report the closest nearest-neighbor. A close observation reveals that an item returned as candidate from a $(K, L)$-parameterized LSH algorithm is sampled with probability $1 - (1 - p^K)^L$ where $p$ is the collision probability of LSH function (Leskovec et al., 2014). The precise form of $p$ is defined

by the LSH family used to build the hash tables. However, the traditional LSH algorithm does not represent a valid probability distribution $\sum_{i=1}^{N} Pr(y_i) \neq 1$. Also, due to the nature of LSH, the sampled candidates are likely to be highly correlated. It turns out that there is a simple, unbiased estimator for the partition function using the samples from the LSH algorithm.

**LSH Sampler and Partition Function Estimator:** Assume there is a set of states $Y = [y_1 \ldots y_N]$. We associate a probability value $p_i$ with each state $y_i$. Here is the description of the sampling process: We flip a Bernoulli coin $m_i$ with probability $p_i$ for each state $y_i$. The sample set $S$ contains all of the states accepted by the Bernoulli sampling process. The probabilities are not required to sum to 1, and the sampling process is allowed to be correlated. Given the sample set $S$, we have an unbiased estimator for the partition function.

$$m_i \sim P(m_i = 1 | p_i) \qquad y_i \in S \iff m_i = 1 \tag{1}$$

**Theorem 2.1.** *Assume that every state $y_i$ has a weight given by $f(y_i)$ with partition function $\sum_{y_i \in Y} f(y) = Z_\theta$. Then we have the following as an unbiased estimator of $Z_\theta$:*

$$Est = \sum_{y_i \in S} \frac{f(y_i)}{p_i} = \sum_{i=1}^{N} \mathbf{1}_{[y_i \in S]} \cdot \frac{f(y_i)}{p_i} \qquad \mathbb{E}[Est] = \sum_{i=1}^{N} \mathbb{E}[\mathbf{1}_{[y_i \in S]}] \cdot \frac{f(y_i)}{p_i} = Z_\theta$$

**Theorem 2.2.** *The variance of the partition function estimator is:*

$$Var[Est] = \sum_{i=1}^{N} \frac{f(y_i)^2}{p_i} - \sum_{i=1}^{N} f(y_i)^2 + \sum_{i \neq j} \frac{f(y_i)f(y_j)}{p_i p_j} \mathrm{Cov}(\mathbf{1}_{[y_i \in S]} \cdot \mathbf{1}_{[y_j \in S]})$$

**Methodology:** Given these observations, we design a fast, scalable approach for estimating the partition function of log-linear models. Here is an overview of our LSH sampling process:

1. During the pre-processing phase, we use randomized hash functions to build hash tables from the weight vectors $\theta_y$ for each class $y \in Y$.

2. For each partition function estimate, we sample a subset of weight vectors $S$ from the hash tables with probability $p = 1 - (1 - \mathcal{M}(\theta_{y_i} \cdot x)^K)^L$, which is monotonic w.r.t. the unnormalized density of the feature vector and the class weight vector

3. For each weight vector $\theta_y$ in the retrieved set $S$, we calculate the probability $p$ of retrieving the element given the query feature vector $x$.

4. The partition function estimate $\hat{Z}_\theta$ is the sum of each unnormalized density $e^{\theta_y \cdot x}$ in the sample set $S$ weighted by the inverse retrieval probability $\frac{1}{p}$.

**Running Time:** The LSH sample size $|S|$ is controlled by two parameters - $K$, the number of bits in the hash fingerprint and $L$, the number of hash tables. The total running time includes the $K \times L$ hash computations, followed by evaluating the formula over the samples returned from the LSH hash tables $O(|S|)$. By increasing $K$ linearly, there is an exponential drop in the sample size. We can fix $K$ such that the expected number of samples in each table is a small constant. In fact, the theory of LSH states that we can ensure a constant number of samples from each hash table when $K = \log n$ where $n$ is the number of states. See (Indyk & Motwani, 1998; Andoni & Indyk, 2004) for more details. Thus, with a small number of hash tables $L$ and $K = \log n$, we can easily obtain a constant sample size, independent of $n$. Therefore, the total computational cost is on the order of $O(\log n)$.

## 3 EXPERIMENTS

We designed the experiment to answer the following three important questions:

1. How accurately does our LSH Sampling approach estimate the partition function?

2. What is the running time of our LSH Sampling approach?

3. How does our LSH sampling approach compare with the alternative approaches in terms of speed and accuracy?

For this experiment, we trained a standard language model on the Penn Tree Bank (PTB) (Marcus et al., 1993) and Text8 (Mikolov et al., 2014) datasets. The output layer is a softmax classifier that predicts the next word in the text using the context vector $x$ generated by the LSTM. Our model is a single layer LSTM with 512 hidden units. The dimensionality of the word embeddings is equal to the number of hidden units. The model is unrolled 20 steps for the back-propagation through time (BPTT). The model is trained with an Adagrad optimizer for 10 epochs with a mini-batch of size 32.

We implemented the following approaches to compare and contrast against our approach:

1. **Uniform Importance Sampling:** An IS estimate where the proposal distribution is a uniform distribution U[0, N]. All samples are weighted equally.

2. **Exact Gumbel:** The Max-Gumbel Trick is used to estimate the partition function. The maximum over all of the states is used for this estimate. (Gumbel & Lieblein, 1954)

3. **MIPS Gumbel:** A MIPS data structure is used to collect a subset of the states efficiently. This subset contains the states that are most likely to have a large inner product with the query. The partition function is estimated via the Max-Gumbel Trick using the subset instead of all the states. (Mussmann & Ermon, 2016; Mussmann et al., 2017)

We took a snapshot of the weights $\theta_y$ for all the words and the context vector $x$, after training the language model for a single epoch. The number of examples in the snapshot is the mini-batch size $\times$ BPTT steps. i.e. (32 examples x 20 steps = 640 total) Using the snapshot, we show how well each approach estimates the partition function by measuring the Mean Absolute Error (MAE). The x-axis is the number of samples used for the partition function estimate. i.e. PTB # samples - [50, 250, 400, 800, 1350] and Text8 # samples - [50, 250, 450, 950, 1800, 3350]. Table 1 shows how the wall-clock time performance scales w.r.t the number of samples.

Figure 1 shows that our LSH estimate is more accurate than the Uniform IS and LSH Gumbel estimates. Exact Gumbel is the most accurate estimator with the lowest MAE for both datasets. MIPS Gumbel is 50% faster than the Exact Gumbel, but its accuracy is significantly worse. Exact Gumbel and MIPS Gumbel are much slower than Uniform IS and LSH by several orders of magnitude. As the number of samples increases, the MAE for the Uniform IS and LSH estimate decreases.

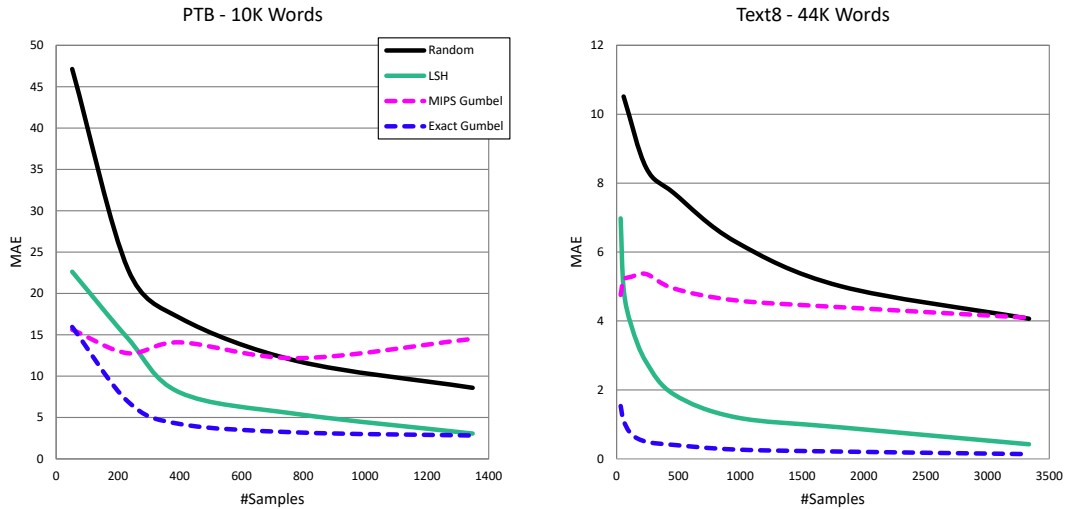

Figure 1: Accuracy of Partition Function Estimate - **(Left)** PTB / **(Right)** Text8

| PTB - 10K words | | | | | Text8 - 44K words | | | | |
|---|---|---|---|---|---|---|---|---|---|
| Samples | Uniform | LSH | Exact Gumbel | MIPS Gumbel | Samples | Uniform | LSH | Exact Gumbel | MIPS Gumbel |
| 50 | 0.10 | 0.19 | 79 | 46 | 50 | 0.13 | 0.23 | 531 | 261 |
| 150 | 0.33 | 0.60 | 249 | 141 | 400 | 0.92 | 1.66 | 3,962 | 1,946 |
| 400 | 0.94 | 1.74 | 690 | 406 | 1500 | 3.41 | 6.14 | 14,687 | 7,253 |
| 1000 | 1.87 | 3.44 | 1,648 | 1,064 | 5000 | 9.69 | 17.40 | 42,035 | 20,669 |

Table 1: Wall-Clock Time performance (seconds) for the Partition Function Estimate

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
