# OpenReview forum: "Scalable Estimation via LSH Samplers (LSS)"
_ICLR.cc/2018/Workshop — Accept_

### Official Review · AnonReviewer2 · 2018-03-09
**A novel sampling based approach to estimate the partition function**

**Rating:** 7
**Confidence:** 4

**Review:**

Estimating the partition function is a challenging problem and new approaches to do this more efficiently are likely to be significant in a number of areas.
Pros
Simple to implement approach
Asymptotic guarantees
Shows good results on limited benchmarks

Cons
motivation is somewhat lacking given that there are many different approaches to partition function estimation (hashing-based, sampling-based, etc.)

Overall, the paper shows promise of solid future work given the initial experimental results and the idea itself.

---

### Official Review · AnonReviewer3 · 2018-03-15
**simple and yet interesting sampling approach for softmax**

**Rating:** 7
**Confidence:** 4

**Review:**

This workshop paper showcases that locality sensitive hashing (LSH), an approximate nearest-neighbor (ANN) approach, could be used as an efficient adaptive sampler. The idea is somewhat simple but very interesting and appealing. Authors should cite works of others on the similar venue. More specifically the following paper:
https://openreview.net/pdf?id=SJ3dBGZ0Z
In the time performance analysis it may be better to represent the speed improvement relative to the Exact Gumble rather than showing the absolute numbers. Also it would be interesting to observe the MAE/speedup trade-off.
Formatting suggestions:
LSH Sampling -> LSH sampling?
IS should be introduced on page 1.
Max-Gumbel -> Gumbel-Max?

---

### Official Review · AnonReviewer1 · 2018-03-16
**Good research direction. Needs better application.**

**Rating:** 4
**Confidence:** 5

**Review:**

Summary:

This paper proposes a new sampling scheme to estimate the partition function by using LSH. The proposed sampler produces samples which are correlated and unnormalized. However authors propose a way to use these samples to derive an unbiased estimator for the partition function. There are some experimental results on estimating the denominator of the softmax function.

My comments:

While the idea of using LSH to estimate the partition function in sub-linear time is good, I feel that the motivating example of larger softmax in the attention mechanism is not very convincing. A simple solution to approximate softmax in the attention mechanism is to find top-k candidates and take soft-max only based on these top-k candidates. Such an approach has been already explored in Rae et al. 2016, and Chandar et al., 2016. In attention-based application, it is often not necessary to estimate the partition function as long as the performance is improved in other ways.

The experimental setup is not very clear. The word embeddings change during training. How do authors update the LSH? Is it the case the authors compare different approaches only using a single snapshot? If so, it is not very useful for this application.

While the problem and research direction is good, I recommend the authors to choose a different application where the solution would be really useful.


References:

Rae et al. 2016, Scaling Memory-Augmented Neural Networks with Sparse Reads and Writes.
Chandar et al., 2016, Hierarchical Memory Networks.

---

### Decision · Program_Chairs · 2018-03-20
**ICLR 2018 Workshop Acceptance Decision**

**Decision:**

Accept

**Comment:**

Congratulations, your paper was accepted to the ICLR workshop.